# Paramagnetic Properties and Moderately RapidConformational Dynamics in the Cobalt(II) Calix[4]arene Complex by NMR

**DOI:** 10.3390/molecules27144668

**Published:** 2022-07-21

**Authors:** Eugeny Nikolaevich Zapolotsky, Sergey Pavlovich Babailov, Gennadiy Alexandrovich Kostin

**Affiliations:** Nikolayev Institute of Inorganic Chemistry SB RAS, 630090 Novosibirsk, Russia; babajlov@niic.nsc.ru (S.P.B.); kostin@niic.nsc.ru (G.A.K.)

**Keywords:** complexes of lanthanides, calix[4]arene, tetraphosphineoxide substituent, molecular dynamics, Gibbs free energy of activation, dynamic NMR

## Abstract

^1^H NMR measurements are reported for the CD_2_Cl_2_/CDCl_3_ solutions of the Co(II) calix[4]arenetetraphosphineoxide complex (**I**). Temperature dependences of the ^1^H NMR spectra of **I** have been analyzed using the line shape analysis, taking into account the temperature variation of paramagnetic chemical shifts, within the frame of the dynamic NMR method. Conformational dynamics of the 2:1 Co(II) calix[4]arene complexes was conditioned by the *pinched cone* ↔*pinched cone* interconversion of **I** (with activation Gibbs energy Δ*G***^≠^**(298K) = 40 ± 3 kJ/mol. Due to substantial temperature dependence of paramagnetic shifts, complex **I** can be used as model compound for designing an NMR thermosensor reagent for local temperature monitoring.

## 1. Introduction

The chemistry of calix[4]arenes is currently very extensive, as it has been intensively developed over the past few decades [1,2,3,4,5,6,7,8,9,10,11]. The nomenclature of the various bulky substituents in the phenolic hydroxyls of calix[4]arenes may already be measured in hundreds of varieties. One of the problems of the physicochemistry of calixarenes is the presence of conformational dynamics associated with the mutual transition between several conformations (in particular, *cone*, *partial cone*, 1,2-*alternate* and 1,3-*alternate* [12]). Various substituents make a significant contribution to the nature of molecular motions, determining the mechanism and, ultimately, their energy parameters [1,6,7,13,14,15,16]. Information of this kind is essential for the directed design of host–guest complexes and other supramolecular structures. The analysis of this kind of conformational transitions is often complicated, and NMR is a method that, in some cases, makes it possible to make certain predictions about the type of dynamics and its energy component.

Earlier complexes of *d*-elements with some calix[n]arenes were characterized by NMR [9,10]. However, paramagnetic complexes of this class have not previously been investigated sufficiently in detail by dynamic NMR (DNMR) [17]. The goal of the present work is a detailed study of the intramolecular conformational dynamics in Co(II) calix[4]arene complexes (**I**) with the use of the ^1^H NMR band-shape analysis. Taking into accountthe data on the temperature dependence of paramagnetic chemical shifts, conformational intramolecular dynamics of complex **I**was assayed. This work is focused on the dynamics of the pinched cone ↔ pinched cone interconversion of the 2:1 Co(II) calix[4]arene complex. In this paper, we propose to use the complete band-shape analysis technique within the framework of the dynamic NMRfor a detailed investigation of the intramolecular dynamics of the paramagnetic Co^2+^complex at different temperatures. In this view, the substantial changes in the paramagnetic chemical shifts depending on the temperature can be taken into account by extrapolating within the Curie law (analogously to 3*d*- and 4*f*-complexes [18,19,20,21]). This technique allows not only to determine the rate constant at room temperature but also to estimate the activation free energy Δ*G^≠^*of conformational dynamics at different temperatures. Such studies on intramolecular dynamics in paramagnetic *d*-complexes in the literature are very scarce.

Taking Co^2+^ as an example, the methodology of paramagnetic *d*-element probe applications for the study of the free-energy variation in chemical exchange processes is discussed. Advantages of this method, compared to DNMR studies of diamagnetic substances, are revealed. In particular, the extension of the range of the NMR-determined rate constants for paramagnetic *d*-element complexes, compared to those of diamagnetic ones, is demonstrated. Lastly, for practical applications, the usage of investigated coordination compounds as a new type of thermometric NMR sensors and paramagnetic probes for in situ temperature control in a solution could be recommended [22,23]. The Co^2+^ complex **I**is considered to bea model system for the design of a thermosensing contrast reagent for MRI diagnostics of cancer and inflammation, and a potential new functional material.

## 2. Results and Discussion

In the ^1^H NMR spectra of **I**, there were peaks corresponding to protons of various H-containing groups in the range from −35 to 35 ppm at the lowest temperature of 173 K, and in the range from 26.0 to −6.4 ppm at the highest temperature of 310 K (Figure 1, Figure 2 and Appendix A). Nine resonances assigned to the complex **I**protons were observed at 310 K. Almost all the resonances were shifted and broadened, indicating the presence of a paramagnetic complex in the solution.

To assign the signals in the NMR spectra, the combined analysis of the integrated signal intensities and paramagnetic chemical shifts was used. The resonances of CH_2_ protons of phosphineoxide moiety were readily determined due to substantial chemical shifts correlating with the distance from the paramagnetic Co(II) cation to the resonating protons [24]. This was observed because the cooperating action of both the contact and dipole shifts to the proton resonances of-Bu_2_P(O)- and -O-CH_2_- groups compactly placed in a limited area relative to the anisotropy axis of the paramagnetic center. Conversely, the signals *t*-Bu and bridged CH_2_ protons underwent the smallest paramagnetic shift (compared to the diamagnetic one), and they were also easily distinguished by the integral intensity and position of the signals.

The assignment of nonequivalent (due to different positions relative to paramagnetic centers) aromatic protons, which are in the negative scale of chemical shifts, was also not difficult. Most of the signals were not in stationary positions, but changed with the temperature (Figure 2, Appendix A) as a result of chemical exchange processes due to conformational dynamics of the complexes. In addition, due to the presence of a paramagnetic center, the signals changed their positions significantly with temperature.

The chemical exchange observed in the NMR spectra manifested itself most clearly at the position of the CH_2_ resonances of phosphine oxide and aromatic protons, since when the conformations of the mobile segments changed, their positions relative to the paramagnetic center changed significantly. In order to observe all separate signals, we tried to use the effect of paramagnetic shifts in NMR spectra of the cobalt complex and extremely low temperatures (173 K for a 1:6 vol. mixture of CDCl_3_ and CD_2_Cl_2_). The combination of these conditions gave us slow (in terms of the NMR time scale) chemical exchange observing clearly on aromatic H_ar_ and CH_2_ resonances. Therefore, a pair of signals (−14.6 and −29.7 ppm) of aromatic protons H_ar_ abruptly disappeared with increasing temperature from 173 to 190 K, but another set of broadened H_ar_ signals (−15.5 and −26.3 ppm at 210 K) appeared, participating in a chemical exchange with each other with a further increase in temperature; as a result of the exchange, one signal remained from this pair (−6.4 ppm at 310 K). It can be assumed that at low temperatures (173 K), one form («**A**») of complex (I) predominates, and upon transition from the region of low temperatures to higher temperatures (190–230 K), the form «**A**» gradually reduced, and another form («**B**») began to appear and then predominate.

It should be noted that the signals of aromatic protons ofform «**A**» at low temperatures (from 173 to 210 K) did not undergo chemical exchange, which was slow on the NMR time scale up to 230 K, and then, the signal was going to broadening (Figure 2). Meanwhile, upon transition to form «**B**» (existing at temperatures from 230 to 310 K), its conformational state was more versatile (which was expressed in the presence of a slow chemical exchange). Interestingly, the «**B**» system was thermodynamically more stable than the «**A**» system during the “defrosting” process (at the temperature increases).

It can be seen in the NMR spectra, that the dynamics described above werereflected on the position of the methyl protons of *tert*-butyl. At the lowest temperature of 173 K, the broad *t*-Bu’ signal belonging to form «**A**» (Appendix A) was shifted upfield to −1.0 ppm, and, at temperature increases, this signal was shifted downfield to 0.7 ppm and became narrow. This means that at high temperatures, the part of *t*-Bu groups belonging to form «**B**» became equidistant relative to the paramagnetic center.

We investigated the band shape of the zero-order ^1^H NMR spectra of complex **I** over a temperature range (*T* = 173−310 K) for both forms in order to evaluate the magnitude of the activation free energy of the molecular dynamics using the temperature dependence of rate constants by Eyring equation (Figure 3).

The determination of the rate constants of the processes and the free energy of activation was similar to the procedure described in the works [21,25] (related analytical expressions are also presented in [17]; Figure 3). The Gibbs free energy for the activation Δ*G^≠^*(298 K)of intramolecular dynamics in both forms was very close to each other and had the values of 40 ± 3 and 37 ± 3 kJ/mol for «**A**» and «**B**» forms, respectively.

The molecular dynamics occurring in calixarene complexes are described by different processes. One of them, and one that has been relatively wellstudied [26], is the *pinched cone*-*pinched cone* interconversion described above. In addition, structural rearrangements associated with transitions between *cone*, *partial cone*, 1,2-*alternate* and 1,3-*alternate* states are possible [26]. At different times and for different compounds, the energy parameters of such conformational transitions were calculated and observed experimentally by NMR and other methods [5,26,27]. The free energy value obtained by us corresponds to that for processes in related complexes, in which the values of Δ*G^≠^*(298 K) are in the range from 25 to 60 kJ/mol [2,4,5,6,26]. The value of the activation energy for different types of conformational dynamics is individual for each compound. The general rule (not without exceptions) is that the potential barrier to *cone-cone* dynamics is higher than the corresponding *alt-cone* transitions. The close values of Δ*G^≠^*(298 K) for «**A**» and «**B**» forms most likely assigned to the *cone-cone* dynamics in both cases (Figure 4).

Previously, coordination equilibria of Co complexes with similar compounds have been studied [16,24], and crystal structures of some complexes have been determined. In contrast, in the crystal structure of complex **I** studied in the investigation, two different coordination modes were detected. One cobalt cation was coordinated by three phosphine oxide centers and located far from the upper rim *t*-Bu groups. Another cobalt cation was coordinated by only one phosphine oxide center (Figure 5).

Taking into account the spectral pattern observed in ^1^H NMR spectra at different temperatures and structural data, we can suppose that the investigated complex underwent the following transformation with the temperature increase (see Figure 1 below):

Unfortunately, in favor of such a hypothesis, there is no way to add more convincing evidence. One can only conclude that, regardless of the coordination equilibrium, the energy barriers difference for the conformational dynamics is not so significant.

Regardless of the coordination equilibrium, the conformational pinched cone ↔ pinched cone dynamics manifests itself in a typical way, and some difference between the corresponding magnetic chemical shifts on temperature in the working range is adequately described by the Curie–Weiss approximation [28,29,30]:*δ* = *a* + *b*/*T*(1)
where *a* and *b* are the parameters of linear regression. We analyzed the linear dependence of the observed chemical shifts on parameter *1*/*T* (see Figure 6 and Appendix A) and found it to be well-described by the Curie–Weiss law.

We also investigated the actual temperature dependence of the chemical shifts on temperature. This was done in order to compare the obtained temperature sensitivities (the temperature derivatives ∂*δ*/∂*T*) with the corresponding data in the literature (when investigated on other nuclei). This maximum value for **I** appeared to be ∂*δ*/∂*T* = 0.16 ppm/K. It should be noted that other complex compounds, which were investigated earlier by ^1^H NMR in both organic and aqueous media, are characterized by the chemical shifts temperature sensitivity in the range from 0 to 1.5 ppm/K [22,29,30]. The obtained results indicate that the coordination compound **I** may be considered as amoderately sensitive ^1^H NMR*d*- element paramagnetic probe.

## 3. Conclusions

^1^H and NMR measurements are reported for the CD_2_Cl_2_/CDCl_3_ solutions of the Co(II) calix[4]arene tetraphosphineoxide complex (**I**). Temperature dependencies of the ^1^H NMR spectra of **I** have been analyzed using the line shape analysis, taking into account the temperature variation of paramagnetic chemical shifts, within the frame of the dynamic NMR method. Two differing forms of complexes (presumably with different coordination modes of cobalt cations) observed in NMR spectra were participated in mutual transition. Conformational dynamics of the complexes were conditioned by the *pinched cone* ↔ *pinched cone* interconversion of Co(II) calix[4]arene complexes with similar activation Gibbs energy Δ*G***^≠^**(298K)**of** about 40± 3kJ/mol. Due to substantial temperature dependence of paramagnetic shifts, complex **I** can be used as a model compound for the design of thermosensor reagent for local temperature monitoring.

## 4. Experimental

Complex [Co_2_(NO_3_)_4_L] (where L = calix[4]arene) was prepared by methods similar to those in [16,24]. Co(NO_3_)_2_·6H_2_O (0.2 mmol) was stirred at room temperature in a solution of 1 (0.1 mmol) in CH_2_Cl_2_ (3–5 mL) until complete dissolving of cobalt nitrate. After evaporation of the mixture to 1–1.5 mL, the violet [Co_2_(NO_3_)_4_***1**]was precipitated by the addition of (C_2_H_5_)_2_O (10–15mL). The complex was well-soluble in CH_2_Cl_2_, CHCl_3_ and acetone and almost insoluble in heptane. A single crystalsuitable for X-ray analysis was prepared by slow diffusion of heptane to the solution of the complex in CH_2_Cl_2_in1–2 days. The complex crystallized as a solvate [Co_2_(NO_3_)_4_*1]·(C_2_H_5_)_2_O, losing solvent molecules into the air. IR(cm^−1^): 792, 810 −δ(NO_3_); 1275, 1300 −ν_s_(ONO_2_); 1479, 1513 −ν_as_(ONO_2_); 1008, 1022 −ν (N=O); 1123 −ν (P=O). The purity of calixarene itself according to CHN analysis was no less than 95%.

The ^1^H NMR spectra were recorded by an Avance-III-500 spectrometer (with an operating frequency of 500 MHz). The solutions were 0.01 M in a 1:6 vol. mixture of CDCl_3_ and CD_2_Cl_2_ for NMR spectroscopy with a tetramethylsilane (TMS) internal standard. The solution temperature was controlled by using a Bruker B-VT temperature controller (accuracy was ±0.3 K). Fourier-transformed spectra were subjected to band-shape analysis on a personal computer (WindowsXP) to obtain rate data for complex **I**. Main NMR acquisition parameter values: NS = 400; SW = 200,000 Hz; PW = 11.2 μs; RD = 1 s; DT = 5 μs; NP = 32,768. Temperature calibration was carried out using ethylene glycol and methanol samples.

The rate constants of the intramolecular dynamics were evaluated by the band-shape method [17] for a two-site exchange (see Appendix A).

Activation dynamics parameters (Δ*G^≠^*,kJ/mol) were found using the Eyring equation:(2)$$k=\kappa \frac{k_{B}T}{h}\exp\left(\frac{\Delta G^{\neq }}{RT}\right)$$
where *κ*is thetransmission coefficient (equal to one),*k*_B_isBoltzmann’s constant and *h* is Planck’s constant. An expression (2) for Δ*G^≠^* evaluation was got after numerical values’ substitution:(3)$$\Delta G^{\neq }=0.0914T\left(10.319+\log\left(\frac{T}{k}\right)\right)$$

*^1^H NMRspectral data*. δ_H_(ppm, 1:6 vol. mixture of CDCl_3_ and CD_2_Cl_2_, 310 K): −6.6 (8H, H aromatic), −1.6 (8H, H aromatic), 0.7 (36H, t-Bu), 2.8 (8H, -**CH_2_**- bridge), 6.3 (24H, -CH_3_), 8.3 (16H, -CH_2_-**CH_2_-**CH_3_), 11.6 (16H, -**CH_2_**-CH_2_-CH_3_), 14.1 (16H, -**CH_2_**-CH_2_-CH_2_**-**CH_3_), 26.0 (O-CH_2_).

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
