# Peer review of "Paramagnetic Properties and Moderately RapidConformational Dynamics in the Cobalt(II) Calix[4]arene Complex by NMR"

_molecules, 2022, doi:10.3390/molecules27144668_

Round 1

Reviewer 1 Report

The authors analyzed Co(II) calix[4]arene tetraphosphineoxide complex (1) by 1H-NMR spectroscopy and found that the paramagnetic chemical shifts of some 1H nuclei were very dependent on the temperature. Like various other calix[4]arenes, the compound (1) also had intramolecular conformational exchange dynamics, which the authors analyzed through 1H peak shapes, and they estimated the exchange rates and activation free energy from Eyring's equation. The value of the activation free energy led the authors to suggest that the compound (1) has pinched cone – pinched cone conformational exchanges. There seem to be few examples of dynamic analyses of such paramagnetic d-element complexes. Furthermore, the coordination structures at low and high temperatures were inferred from the peak shapes of the t-butyl group and the crystal structure of a similar Co(II)-coordinating compound. The authors believe that (1) is suitable for a sensor that measures the actual temperature in an NMR sample, proposing that it could also be used in MRI observation, for example, to identify cancer and inflammation parts, which have higher temperatures from others.

Since I specialize in NMR of biological macromolecules, I well understand the NMR analysis parts of the paper. However, I am not familiar with calix[4]arenes, so that I am afraid that I am not qualified to judge well whether the discussion and arguments that take into consideration the past cases on similar substances are correct. Overall, I feel that the manuscript seems to contain more speculation than the experimental data can draw.

One thing I would like to ask the authors is whether they could not have done the NMR analyses using 2D 1H/13C spectra. In one dimensional spectra, the baseline distortion and peak overlaps are likely to create large errors, especially when estimating the molar ratio of isomers from the peak shapes and area. On the other hand, I expect that two-dimensional spectra would generally allow easier and more precise analyses because peaks are more separated. Furthermore, heteronuclei, 15N and 13C, are more tolerant to paramagnetic relaxation enhancement than 1H, so that those closer to Co(II) can be observed.

Minor points:

In Line 34 in Page 1, an abbreviation “DNMR” appears for the first time here. Seven lines below it, the term “dynamic NMR” is written.

In the chemical structure of compound (1) in Figure 1, is the position of the CH2 bridges on the aromatic ring wrong?

Please specify the temperature for each 1D NMR spectrum of Figures S1 and S2.

Author Response

Dear Reviewer!

There is our answers on yours comments.

The authors analyzed Co(II) calix[4]arene tetraphosphineoxide complex (1) by 1H-NMR spectroscopy and found that the paramagnetic chemical shifts of some 1H nuclei were very dependent on the temperature. Like various other calix[4]arenes, the compound (1) also had intramolecular conformational exchange dynamics, which the authors analyzed through 1H peak shapes, and they estimated the exchange rates and activation free energy from Eyring's equation. The value of the activation free energy led the authors to suggest that the compound (1) has pinched cone – pinched cone conformational exchanges. There seem to be few examples of dynamic analyses of such paramagnetic d-element complexes. Furthermore, the coordination structures at low and high temperatures were inferred from the peak shapes of the t-butyl group and the crystal structure of a similar Co(II)-coordinating compound. The authors believe that (1) is suitable for a sensor that measures the actual temperature in an NMR sample, proposing that it could also be used in MRI observation, for example, to identify cancer and inflammation parts, which have higher temperatures from others.

Since I specialize in NMR of biological macromolecules, I well understand the NMR analysis parts of the paper. However, I am not familiar with calix[4]arenes, so that I am afraid that I am not qualified to judge well whether the discussion and arguments that take into consideration the past cases on similar substances are correct. Overall, I feel that the manuscript seems to contain more speculation than the experimental data can draw.

One thing I would like to ask the authors is whether they could not have done the NMR analyses using 2D 1H/13C spectra. In one dimensional spectra, the baseline distortion and peak overlaps are likely to create large errors, especially when estimating the molar ratio of isomers from the peak shapes and area. On the other hand, I expect that two-dimensional spectra would generally allow easier and more precise analyses because peaks are more separated. Furthermore, heteronuclei, 15N and 13C, are more tolerant to paramagnetic relaxation enhancement than 1H, so that those closer to Co(II) can be observed.

We agree with the remark. 2D NMR techniques are quite time consuming. The application of these techniques in the case of paramagnetic systems requires highly skilled operators working on the NMR spectrometer. There are difficulties with the selection of mixing time (mixing time) and phase correction. There are few examples of real-life applications of 2D NMR (in particular 1H-1H NOE) in the literature. We have previously, for example, been able to successfully apply 2D EXSY to paramagnetic praseodymium complexes (published in the journal Inorganic Chemistry). In this study, we made attempts to obtain spectra of 1H-1H NOE. However, the spectra of satisfactory quality could not be obtained. Perhaps this was due to the presence of molecular dynamics.

In this regard, in this work we present only 1D NMR data.

Minor points:

In Line 34 in Page 1, an abbreviation “DNMR” appears for the first time here. Seven lines below it, the term “dynamic NMR” is written.

Answer: Fixed 

In the chemical structure of compound (1) in Figure 1, is the position of the CH2 bridges on the aromatic ring wrong?

Answer: Fixed

Please specify the temperature for each 1D NMR spectrum of Figures S1 and S2.

Answer: Fixed

Reviewer 2 Report

Zapolotsky et al. report the temperature-dependent 1D 1H NMR spectra of 2:1 Co2+:calix[4]arene complex. They analyze the NMR line shapes and extract thermodynamic values for the transition between two exchanging conformations which they attribute to different coordination modes of the Co2+ ion. The model is loosely based on observation and more on previous published structures and experiments which weakens the authors conclusions. Some critical information is missing to assess the work’s soundness. A lot more advanced experiments and analysis is necessary to properly support the conclusions. The manuscript would benefit from careful editing and clarification.

L34 – if DNMR (dynamic NMR?) stands for something specific, please expand and cite a reference for the framework

L108 – there is missing explanation for how the rates were extracted from chemical shift perturbation and what signals were fitted. Please detail the process, especially since the only reference for the framework is from a 1982 book (maybe in the Supplement). Why are there only 4 points in the Eyring plot? If I interpret correctly, A and B correspond to the two species seen in low temperature spectra. It seems like there is significant contribution from intermediate exchange, was this taken in account in the fit equations? What about fitting the shape of the signal seen at higher temperature spectra?

L117 – I do not think the interconversion was described above. It would be helpful to detail it here or in Figure 4 caption.

L138 – Scheme 1

L141 – I am not convinced about the explanation for tBu signals. The change in temperature alone could explain the observations. The authors should have a control experiment with diamagnetic ion.

L153 – Was the figure taken from a published article?

L155 – Expand VT NMR

L165 – I do not agree. Using 2D NMR at lowest and highest temperature, the authors should be readily able to derive structural models based on 1H-1H NOE and distance restraints from paramagnetic shifts.

L170 – which observed chemical shifts?

Author Response

Dear Reviewer!

There is our answers on yours comments.

Zapolotsky et al. report the temperature-dependent 1D 1H NMR spectra of 2:1 Co2+:calix[4]arene complex. They analyze the NMR line shapes and extract thermodynamic values for the transition between two exchanging conformations which they attribute to different coordination modes of the Co2+ ion. The model is loosely based on observation and more on previous published structures and experiments which weakens the authors conclusions. Some critical information is missing to assess the work’s soundness. A lot more advanced experiments and analysis is necessary to properly support the conclusions. The manuscript would benefit from careful editing and clarification.

L34 – if DNMR (dynamic NMR?) stands for something specific, please expand and cite a reference for the framework

Answer: Fixed, the method used is added to SupMat.

L108 – there is missing explanation for how the rates were extracted from chemical shift perturbation and what signals were fitted. Please detail the process, especially since the only reference for the framework is from a 1982 book (maybe in the Supplement). Why are there only 4 points in the Eyring plot? If I interpret correctly, A and B correspond to the two species seen in low temperature spectra. It seems like there is significant contribution from intermediate exchange, was this taken in account in the fit equations? What about fitting the shape of the signal seen at higher temperature spectra?

Answer: In the book of J. Sandstrom "Dynamic NMR spectroscopy", in our opinion, methods for calculating rate constants and activation parameters are excellently and in detail described. For clarity, in SUP MAT we present the methodology for calculating the rate constants that we used. In our work, we used approximate methods to find the rate constants (hence the small number of points on the Eyring plot). In Figure 2, on the temperature spectra, a special dotted line indicates which signals undergo chemical exchange.

L117 – I do not think the interconversion was described above. It would be helpful to detail it here or in Figure 4 caption.

Answer:  The diagram in Fig. 4 describes the standard diadynamic process for calixarenes - pinched cone ↔ pinched cone interconversion. In this part, we describe the observations that come with the spectra, and the interpretation is given later in the text. The main difficulty in processing spectral information was that the conformational transition and thermodynamic equilibrium occur simultaneously. However, by estimating the activation energy and analyzing the spectral patterns, we were able to satisfactorily describe the system in these terms. We agree that our explanation can be hard to read and have improved the text for better understanding.

L138 – Scheme 1

Answer: Fixed

L141 – I am not convinced about the explanation for tBu signals. The change in temperature alone could explain the observations. The authors should have a control experiment with diamagnetic ion.

Answer. We believe that our interpretation of the behavior of the tBu signals is quite satisfactory. Both conformational dynamics and the thermodynamic transition between forms A and B are traced. For clarity, we supplemented Fig S2 with the necessary notation and corrected the text of the article in the direction of greater clarity.

L153 – Was the figure taken from a published article?

Answer: All the figures have original design excluding fig 5, getting from  [10.1007/s10847-010-9755-y] (Ref.[16])

L155 – Expand VT NMR

Answer: Fixed

L165 – I do not agree. Using 2D NMR at lowest and highest temperature, the authors should be readily able to derive structural models based on 1H-1H NOE and distance restraints from paramagnetic shifts.

Answer: We agree with the remark. 2D NMR techniques are quite time consuming. The application of these techniques in the case of paramagnetic systems requires highly skilled operators working on the NMR spectrometer. There are difficulties with the selection of mixing time (mixing time) and phase correction. There are few examples of real-life applications of 2D NMR (in particular 1H-1H NOE) in the literature. We have previously, for example, been able to successfully apply 2D EXSY to paramagnetic praseodymium complexes (published in the journal Inorganic Chemistry). In this study, we made attempts to obtain spectra of 1H-1H NOE. However, the spectra of satisfactory quality could not be obtained. Perhaps this was due to the presence of molecular dynamics.

In this regard, in this work we present only 1D NMR data.

L170 – which observed chemical shifts?

Answer: Fixed

Reviewer 3 Report

Zapolotsky et al. investigate the conformational dynamics of a cobalt(II) calixarene complex at varying temperature and analyze the changes of paramagnetic NMR resonances. The work appears sound and experiments well conducted. I believe the readability of the manuscript could be much improved adding some more explanations. Furthermore, an overall grammar/spell check is required.

What is the presumed spin state of Co(II) in the investigated complex?

What is the evidence for a 2:1 Co:calixarene stoichiometry?

Resonance assignments are based solely on 1D spectra analysis, or have they been confirmed by 2D spectroscopy? Please clarify.

“slow (in terms of the NMR time scale) chemical exchange”: please provide estimated exchange rates.

Experimental section: preparation procedure, even if not original, should be described in brief. What was the purity of the calixarene?

The investigated complex is proposed as a model system for developing NMR thermometric sensors for in situ applications. However, the compound was solubilized in organic solvent, please explain the adaptations required for use in biological media.

Please provide main NMR acquisition parameter values.

Band shape analysis: the procedure could be succinctly explained and possibly show some results (e.g. simulated resonances?).

Eq. 3: please define a and b.

Fig 1, 2. X axis units are unclear.

Author Response

Dear Reviewer!

There is our answer on yours comments

Zapolotsky et al. investigate the conformational dynamics of a cobalt(II) calixarene complex at varying temperature and analyze the changes of paramagnetic NMR resonances. The work appears sound and experiments well conducted. I believe the readability of the manuscript could be much improved adding some more explanations. Furthermore, an overall grammar/spell check is required.

 What is the presumed spin state of Co(II) in the investigated complex?

Answer: For octahedral Co(2+) complexes the low-spin state t2g6eg1 is most preferable.

 What is the evidence for a 2:1 Co:calixarene stoichiometry?

 Answer: From X-ray data, 2:1 Co:calixarene stoichiometry was assigned. At high resolution NMR conditions, no change in stoichiometry was observed.  Also complex was described by elemental analysis. For For C80H132P4O20Со2N4 Calc., %:            С – 56.1 Н – 7.71 N – 3.2 Found, %: C – 56.2 H – 7.78 N – 3.19

Resonance assignments are based solely on 1D spectra analysis, or have they been confirmed by 2D spectroscopy? Please clarify.

Answer: We agree with the remark. 2D NMR techniques are quite time consuming. The application of these techniques in the case of paramagnetic systems requires highly skilled operators working on the NMR spectrometer. There are difficulties with the selection of mixing time (mixing time) and phase correction. There are few examples of real-life applications of 2D NMR (in particular 1H-1H NOE) in the literature. We have previously, for example, been able to successfully apply 2D EXSY to paramagnetic praseodymium complexes (published in the journal Inorganic Chemistry). In this study, we made attempts to obtain spectra of 1H-1H NOE. However, the spectra of satisfactory quality could not be obtained. Perhaps this was due to the presence of molecular dynamics.

In this regard, in this work we present only 1D NMR data.

“slow (in terms of the NMR time scale) chemical exchange”: please provide estimated exchange rates.

Answer: The rate constants depending on the temperature were in the range of 103-104 s-1. The values of the constants are easy to obtain from the graphs in Fig. 3, so we did not give numerical values. 

Experimental section: preparation procedure, even if not original, should be described in brief. What was the purity of the calixarene?

Answer: The synthesis of complex [Co2(NO3)4×1] is added to experimental part. Со(NO3)2×6H2O (0.2 mmol) was stirred at room temperature in solution of 1 (0.1 mmol) in CH2Cl2 (3-5 ml) until complete dissolving of cobalt nitrate. After evaporation of the mixture to 1-1.5 ml the violet [Co2(NO3)4*1]  was precipitated by addition of (C2H5)2O (10-15 ml). The complex is well soluble in CH2Cl2, CHCl3 and acetone and almost insoluble in heptane. Single crystal  suitable for X-ray analysis was prepared by slow diffusion of heptane to the solution of complex in CH2Cl2 during 1-2 days. The complex crystallizes as solvate [Сo2(NO3)4*1]×(C2H5)2O losing solvent molecules on air. IR(cm-1): 792, 810 - d(NO3); 1275, 1300 - ns(ONO2); 1479, 1513 - nas(ONO2); 1008, 1022 - n (N=O); 1123 - n (Р=O). The purity of calixarene itself according to CHN analysis was no less than 95 %.

The investigated complex is proposed as a model system for developing NMR thermometric sensors for in situ applications. However, the compound was solubilized in organic solvent, please explain the adaptations required for use in biological media.

Answer: Currently, the search for model systems is not limited to water-soluble complexes. As an example, earlier (10.1016/j.sna.2021.112933) we also studied complexes of Ln with phthalocyanines, soluble in organics, solubilized in water with the help of surfactants. as a thermosensory reagent.

 Please provide main NMR acquisition parameter values.

Answer: Main NMR acquisition parameter values: NS= 400; SW= 200000 Hz; PW= 11.2 usec; RD= 1 sec; DT= 5 usec; NP= 32768.

Band shape analysis: the procedure could be succinctly explained and possibly show some results (e.g. simulated resonances?).

Answer:  The band shape analysis we used is added to SupMat. We used simplified method for rate constant evaluation described in book of J. Sandstrom  «Dynamic NMR spectroscopy».  

 Eq. 3: please define a and b.

Answer: Fixed

 Fig 1, 2. X axis units are unclear.

Answer: Fixed

Round 2

Reviewer 2 Report

I would like to thank the authors for answering my questions. The manuscript greatly improved. However they did miss some critical points that should be addressed before publication. One of them being the absence of a proper negative control, I can ever so strongly encourage the authors to record 1D NMR on a diamagnetic equivalent of their sample.

L108 - Thank you for clarifying the rate extraction process. You still did not consider the case of intermediate exchange on the NMR timescale. Even a small proportion of intermediate exchange can significantly affect the rates and since you use absolute DG values to draw your conclusions, this is important. Please fit the rates accordingly.

Author Response

Point 1:. I would like to thank the authors for answering my questions. The manuscript greatly improved. However they did miss some critical points that should be addressed before publication. One of them being the absence of a proper negative control, I can ever so strongly encourage the authors to record 1D NMR on a diamagnetic equivalent of their sample.

Response 1: Partially agree with the remark, but with a reservation. As we found out, the Co-complex exhibits interesting properties - in addition to conformational dynamics, there are also thermodynamic transitions, which, as we assumed, reflect the process of rearrangement of the coordination polyhedron. Is it safe to assume that some diamagnetic equivalent will be structurally similar to the paramagnetic one and will have a similar type of dynamics? We think that individual differences between compounds will be quite large, and ignoring this may lead to incorrect generalizations.

Point 2:.  L108 - Thank you for clarifying the rate extraction process. You still did not consider the case of intermediate exchange on the NMR timescale. Even a small proportion of intermediate exchange can significantly affect the rates and since you use absolute DG values to draw your conclusions, this is important. Please fit the rates accordingly.

Response 2:  Partly agree with the comment. We use the intermediate exchange formalism for paramagnetic systems if we are sure that we have correctly estimated the position of the coalescence point in the temperature range. Due to the large spectral splitting and intrinsic broadening of the signals, it becomes difficult to clearly fix the coalescence point (because of the smearing of the peaks), hence the large errors. In our case, the numerical estimates for the intermediate exchange rates agreed satisfactorily with those for the slow and fast exchange rates, but we did not include them because we were not sure of their quality (for the reasons described above). For clarity, we added the formula for calculating the rate of intermediate exchange (at the coalescence point) to the methodological part (Sup. Mat.).

As for the correctness of the dG≠ estimate, J.Sandstrom in his book analyzes the errors in detail and shows that, unlike dH≠ and dS≠, the dG≠ parameter is not overly sensitive to errors in the original data. Thus, even approximate methods for finding the values ​​of the rate constants k(T) give satisfactory estimates of dG≠ ± units kJ/mol.